# MIBP-Cert: Certified Training against Data Perturbations with Mixed-Integer Bilinear Programs

**Tobias Lorenz**
CISPA Helmholtz Center
for Information Security
Saarbrücken, Germany
`tobias.lorenz@cispa.de`

**Marta Kwiatkowska**
Department of Computer Science
University of Oxford
Oxford, UK
`marta.kwiatkowska@cs.ox.ac.uk`

**Mario Fritz**
CISPA Helmholtz Center
for Information Security
Saarbrücken, Germany
`fritz@cispa.de`

## Abstract

Data errors, corruptions, and poisoning attacks during training pose a major threat to the reliability of modern AI systems. While extensive effort has gone into empirical mitigations, the evolving nature of attacks and the complexity of data require a more principled, provable approach to robustly learn on such data—and to understand how perturbations influence the final model. Hence, we introduce MIBP-Cert, a novel certification method based on mixed-integer bilinear programming (MIBP) that computes sound, deterministic bounds to provide provable robustness even under complex threat models. By computing the set of parameters reachable through perturbed or manipulated data, we can predict all possible outcomes and guarantee robustness. To make solving this optimization problem tractable, we propose a novel relaxation scheme that bounds each training step without sacrificing soundness. We demonstrate the applicability of our approach to continuous and discrete data, as well as different threat models—including complex ones that were previously out of reach.

## 1 Introduction

Data poisoning attacks are among the most significant threats to the integrity of machine learning models. Attackers can exert far-reaching influence by targeting systems in sensitive domains such as finance, healthcare, or autonomous decision-making. These attacks inject malicious data into the training process, causing models to make incorrect or harmful predictions after deployment. Government agencies across the US and Europe have recognized poisoning as one of the fundamental threats to AI systems and highlight the need for robust defenses in their respective reports [9, 30] and legislation [10].

To counter this threat, many empirical defenses have been proposed [7], relying on data filtering, robust training, or heuristic inspection to mitigate the effect of adversarial data. However, these methods offer no formal guarantees: they may reduce the harm of specific perturbations but cannot ensure robustness in general. As a result, increasingly sophisticated poisoning attacks have been developed to bypass these defenses [29].

39th Conference on Neural Information Processing Systems (NeurIPS 2025).

| Property | FullCert [20] | Sosnin et al. [28] | BagFlip [37] | MIBP-Cert (Ours) |
|---|---|---|---|---|
| Deterministic certificates | ✓ | ✓ | ✗ | ✓ |
| $\ell_0$-norm perturbations | ✗ | (✓)[†] | ✓ | (✓)[†] |
| $\ell_\infty$-norm perturbations | ✓ | ✓ | ✗ | ✓ |
| Complex perturbations | ✗ | ✗ | ✗ | ✓ |
| Real-valued features | ✓ | ✓ | ✗ | ✓ |
| Categorical features | ✗ | ✗ | ✓ | ✓ |
| Stable bounds during training | ✗ | ✗ | – | ✓ |

[†] Support for $\ell_0$ when training on the full dataset in a single batch.

Table 1: Comparison of certified training methods. Only MIBP-Cert combines deterministic certificates, mixed-integer constraints, and support for both real-valued and categorical features.

A recent line of work shifts toward *provable guarantees* that bound the influence of training-time perturbations on model predictions [20, 28]. These methods adapt test-time certifiers [12, 36] to the training process, using interval or polyhedral approximations to track parameter evolution under bounded perturbations. They compute real-valued bounds on model parameters throughout training, constraining the effect of adversarial data. These are important first steps toward certifiable robustness, offering deterministic guarantees under well-defined threat models.

However, while these types of bounds have been shown to be a good trade-off between precision and computational efficiency for test-time certification [19], the significant over-approximations of polyhedral constraints limit the method's stability for training. Lorenz et al. [20] show that interval bounds can cause training to diverge due to compounding errors.

We address this limitation by proposing MIBP-Cert, a precise certification method based on solving mixed-integer bilinear programs (MIBPs). Rather than approximating training dynamics layer by layer, we frame certified training as an exact optimization problem in parameter space. To keep the method tractable, we cut the computational graph after each parameter update and compute bounds iteration-wise. The resulting bounds remain sound and avoid the divergence issues of prior methods. In addition to tighter bounds, our formulation supports more flexible and expressive threat models, including complex conditional constraints and structured perturbations beyond $\ell_p$-balls. Our experimental evaluation confirms the theoretical advantages of MIBP-Cert, showing improved stability and higher certified accuracy for larger perturbations compared to prior work.

To summarize, our main contributions are:

- **New formulation.** We present the first formulation that casts training-time robustness certification as a mixed-integer bilinear program, enabling a different solution path.

- **Theoretical stability.** We show that our approach addresses fundamental convergence issues in existing methods.

- **Expanded scope.** We enable certification of perturbation types that no prior method can handle.

- **Precision gains.** We achieve tighter bounds, leading to certified accuracy improvements, especially in challenging large-perturbation regimes.

The implementation of our method is available at `https://github.com/t-lorenz/MIBP-Cert`.

## 2   Related Work

There are two major lines of work on certified defenses against training-time attacks: *probabilistic ensemble-based methods* and *deterministic bound-based* methods.

**Probabilistic Certificates by Ensemble-Based Methods.** Ensemble-based methods are typically based on either bagging or randomized smoothing. Wang et al. [31], Rosenfeld et al. [26], and Weber et al. [34] extend Randomized Smoothing [8] to training-time attacks. While Wang et al. [31] and Weber et al. [34] compute probabilistic guarantees against $\ell_0$ and $\ell_2$-norm adversaries respectively, Cohen et al. [8] provide these guarantees against label flipping attacks.

A similar line of work provides probabilistic guarantees against training-time attacks using bagging. Jia et al. [15] find that bagging's data subsampling shows intrinsic robustness to poisoning. Wang et al. [33] enhance the robustness guarantees with advanced sampling strategies, and Zhang et al. [37] adapt this approach for backdoor attacks with triggers. Some studies explore different threat models, including temporal aspects [32] and dynamic attacks [5]. Recent work [18] introduces a deterministic bagging method. In contrast, our work advances the state-of-the-art in deterministic certification as well as complex perturbation models.

**Deterministic Certificates by Bound-Based Methods.** In contrast to these probabilistic certificates by ensemble-based sampling methods, Lorenz et al. [20] and Sosnin et al. [28] propose to compute sound deterministic bounds of the model's parameters during training. They define a polytope of allowed perturbations in input space and propagate it through the forward and backward passes during training. By over-approximating the reachable set with polytopes along the way, they compute sound worst-case bounds for the model's gradients. Using these bounds, the model parameters can be updated with sound upper and lower bounds, guaranteeing that all possible parameters resulting from the data perturbations lie within these bounds. Lorenz et al. [20] use intervals to represent these polytopes and extend the approach to also include test-time perturbations. Sosnin et al. [28] use a combination of interval and linear bounds. Both certify robustness to $\ell_\infty$-norm perturbations (i.e., clean-label poisoning), and Sosnin et al. [28] additionally limit the number of data points that can be perturbed. In contrast, we provide *tighter* relaxations by formulating deterministic certification of training in a mixed-integer bilinear program that allows more *stable bounds* and enables *complex perturbations* with both *real-valued and categorical features* (Table 1).

## 3 Certified Training using Mixed-Integer Bilinear Programming

The goal of training certification is to rigorously bound the error that perturbations to the training data can introduce in the final model. This is a fundamental challenge in certifying robustness to data poisoning: we seek guarantees that, under a specified threat model, bound the effect on the trained model and in turn can be used to guarantee correct behavior.

Unfortunately, computing tight bounds on the final model for even small neural networks is infeasible in general. It has been shown to be a $\Sigma_2^P$-hard problem [23], meaning that to make certification tractable, we must settle for over-approximations. The challenge is to construct approximations that (i) are sound, (ii) introduce as little slack as possible, and (iii) remain computationally feasible.

Prior certified training approaches [20, 28] address this by using Interval Bound Propagation (IBP) to derive parameter bounds. While sound, IBP suffers from two key limitations. First, it fails to preserve input-output dependencies, leading to compounding over-approximations even for simple operations. Second, the bounds grow monotonically over training due to subtraction in the parameter update step. As shown by Lorenz et al. [20], this often causes training to diverge entirely.

These issues motivate the core design of our method: we use mixed-integer bilinear programming (MIBP) to compute *exact bounds for a single training step*—including the forward pass, loss, backward pass, and parameter updates. In principle, we could chain these exact encodings over the full training; in practice, this would be computationally prohibitive.

Instead, we relax only the parameter space at each step, bounding the possible parameter configurations resulting from any permitted training data perturbation. This avoids divergence while preserving exact reasoning within each training step. The result is a tractable certification scheme with precise per-step guarantees.

The remainder of this section formally defines our threat model (Section 3.1), demonstrates the approach on an illustrative example (Section 3.2), describes the MIBP formulation for parameter bounds (Section 3.3), and presents the implementation (Section 3.4).

### 3.1 Formal Threat Model and Certificate Definition

Certification provides guarantees that a desired postcondition (e.g., correctness) holds for all inputs satisfying a specified precondition (e.g., bounded perturbations). In our data poisoning setting, the precondition formalizes the adversary's power to perturb the training dataset, and the postcondition encodes the property we aim to verify after training, e.g., correct classification.

**Precondition | Threat Model.** We consider perturbations of the original training dataset $D = \{(x_i, y_i)\}_{i=1}^n$ to a modified dataset $D' = \{(x_i', y_i')\}_{i=1}^n$, subject to constraints. This permits modeling standard $\ell_p$-norm perturbations, as well as rich, dataset-wide threat models that couple samples, features, and labels. While we model perturbations as an adversarial game, they can also have non-malicious, natural causes, such as measurement errors or data biases.

**Definition 3.1** (Family of Datasets). Let $\mathcal{X} \subseteq \mathbb{R}^d$ and $\mathcal{Y}$ denote the input and label domains. The adversary's feasible perturbations are described by a family of datasets:

$$\mathcal{D} := \{D' = \{(x_i', y_i')\}_{i=1}^n \mid x_i' \in \mathcal{X}, \ y_i' \in \mathcal{Y}, \ \mathcal{C}(D, D')\}, \tag{1}$$

where $\mathcal{C}(D, D')$ is a constraint over the original and perturbed datasets.

**Constraint Syntax.** The constraint set $\mathcal{C}(D, D')$ may include arbitrary linear, bilinear, logical, and integer constraints over (1) original and perturbed inputs $x_i$, $x_i'$, (2) original and perturbed labels $y_i$, $y_i'$, (3) auxiliary variables $z_j$, and (4) global variables (e.g., number of modified points). Possible instantiations include (1) pointwise $\ell_\infty$ bounds, e.g., $\|x_i' - x_i\|_\infty \leq \epsilon$; (2) sparsity constraints, e.g., $\sum_{j=1}^m z_j \leq k$ with $z_j \geq \mathbb{I}[x_i[j]' \neq x_i[j]]$; (3) monotonic constraints, e.g., $x_i'[j] \geq x_i[j]$; and (4) class-conditional constraints, e.g., $y_i = c \Rightarrow x_i' = x_i$. This formulation strictly generalizes threat models used in prior certified training work [20, 28, 37].

**Postcondition.** The postcondition is the property we aim to verify for all models trained on permissible datasets. In general, this may include functional correctness, abstention, fairness, or task-specific safety properties. While our method supports arbitrary postconditions that can be encoded as MIBP, we follow prior work and focus on classification correctness for our experiments.

**Definition 3.2** (Certificate). Given a test input $(x, y)$, initial parameters $\theta_0$, and training algorithm $A$, a certificate guarantees:

$$f_{\theta'}(x) = y \quad \forall D' \in \mathcal{D} \ \wedge \ \theta' = A(D', \theta_0). \tag{2}$$

**Parameter Bounds.** Verifying if Eq. (2) holds directly is typically intractable. Instead, we bound the space of parameters reachable by the training algorithm under the threat model:

**Definition 3.3** (Parameter Bounds). We define bounds $[\underline{\theta}, \overline{\theta}]$ such that

$$\theta' = A(D', \theta_0) \Rightarrow \theta' \in [\underline{\theta}, \overline{\theta}] \quad \forall D' \in \mathcal{D}. \tag{3}$$

This yields a sufficient condition for certification:

**Proposition 3.4.** *If $f_{\theta'}(x) = y$ for all $\theta' \in [\underline{\theta}, \overline{\theta}]$, then the certificate condition in Definition 3.2 is satisfied.*

*Proof.* Follows directly from Definition 3.3 by substitution. $\square$

**Iterative Relaxation.** Because training unfolds across multiple iterations, we refine parameter bounds at each step. Let $A_i$ denote the $i$-th update step. Then we compute bounds recursively as:

$$\underline{\theta}_i \leq A_i(D', \theta') \leq \overline{\theta}_i \quad \forall D' \in \mathcal{D}, \ \theta' \in [\underline{\theta}_{i-1}, \overline{\theta}_{i-1}], \tag{4}$$

with $\underline{\theta}_0 = \theta_0 = \overline{\theta}_0$ by initialization. Each step propagates bounds forward while maintaining soundness with respect to $\mathcal{D}$. This recursive strategy enables certifiably robust training under expressive constraints, especially well-suited for tabular data.

## 3.2 Illustrative Example

To illustrate how MIBP-Cert captures the training process, we walk through a simple fully connected model with a single training step on a simplified example (Fig. 1). This helps build intuition for the constraint sets described in the next section. The example has two inputs, $x_1$ and $x_2$. The first operation is a fully connected layer (without bias) with weights $w_{11}$, $w_{12}$, $w_{21}$, and $w_{22}$. This linear layer is followed by a ReLU non-linearity and a second fully connected layer with weights $w_5$ and $w_6$. We use $(x_1 = 1, x_2 = 1)$ as an example input with target label $t = 1$. We set the perturbations $\epsilon = 1$, which leads to upper and lower bounds of $[0, 2]$ for both inputs. In the following, we highlight all equations that belong to the optimization problem in teal. The full optimization problem with all constraints is in Appendix A.

**Forward Pass.** The first step is to encode the forward pass as constraints of the optimization problem. We start with the input by defining $x_1$ and $x_2$ as variables of the optimization problem. By the definition of the threat model, their constraints are $0 \leq x_1 \leq 2$ and $0 \leq x_2 \leq 2$.

For the fully connected layer, we encode the variables $x_3$ and $x_4$ with respect to $x_1$ and $x_2$. To do so, we need the parameters $w_{ij}$ as variables, which are set to their initial values $w_{11} = 1$, $w_{12} = 1$, $w_{21} = 1$, and $w_{22} = -1$. One might be tempted to substitute these variables with their values to simplify the optimization problem. However, this would cause information loss, which would decrease the precision of the final solution. This effect will be amplified in later rounds where $w_{ij}$ are no longer single values but intervals with upper and lower bounds. We add the constraints $x_3 = w_{11}x_1 + w_{21}x_2$ and $x_4 = w_{12}x_1 + w_{22}x_2$ to the optimization problem. Since we multiply two variables by each other, the problem becomes bilinear, which is one of the reasons we require a bilinear optimization problem.

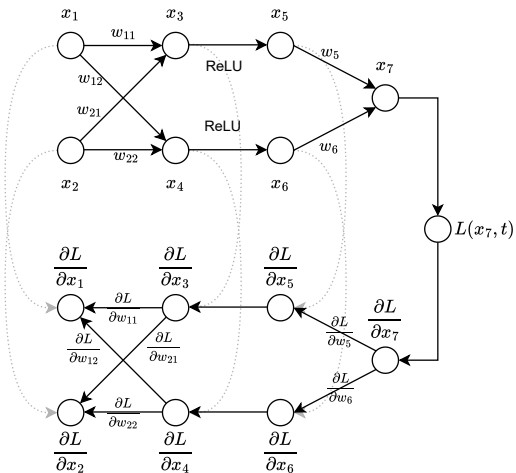

Figure 1: Illustration of a single forward and backward pass for a simplified model.

The ReLU layer can be directly encoded as a piecewise linear constraint: $x_5 = \max(0, x_3)$ or, equivalently, $x_5 = x_3$ if $x_3 > 0$, and $x_5 = 0$ otherwise. $x_6$ is defined accordingly. ReLUs are the main reason why we require mixed-integer programming, as they allow us to encode piecewise linear constraints using binary decision variables. $x_7$ is a linear combination of the two outputs: $x_7 = w_5x_5 + w_6x_6$ with $w_5 = -1$ and $w_6 = 1$. At this stage, we can already see that MIBP-Cert's bounds are more precise: Computing the bounds for $x_7$ using FullCert, we get $-4 \leq x_7 \leq 2$, while solving the optimization problem gives $-4 \leq x_7 \leq 0$ with a tighter upper bound.

**Loss.** We also encode the loss function as constraints. Here, we use the hinge loss $L(x_7, t) = \max(0, 1 - tx_7) = \max(0, 1 - x_7)$ in this example because it is a piecewise linear function and therefore can be exactly encoded, analogous to ReLUs. General losses can be supported by bounding the function with piecewise linear bounds.

**Backward Pass.** For the backward pass, we need to compute the loss gradient for each parameter using the chain rule. It starts with the last layer, which is $\frac{\partial L}{\partial x_7} = -1$ if $x_7 \leq 1$, $0$ otherwise. This is also a piecewise linear function and can be encoded as a constraint to the optimization problem.

The gradients for the linear layer $x_7 = w_5x_5 + w_6x_6$ can be determined using the chain rule: $\frac{\partial L}{\partial w_5} = x_5 \frac{\partial L}{\partial x_7}$ and $\frac{\partial L}{\partial x_5} = w_5 \frac{\partial L}{\partial x_7}$, with corresponding expressions for $x_6$ and $w_6$. Given that the outer gradient $\frac{\partial L}{\partial x_7}$, as well as $x_5$ and $w_5$, are variables, this backward propagation leads to bilinear constraints. The derivatives of ReLU are piecewise linear, resulting in $\frac{\partial L}{\partial x_3} = 0$ if $x_3 \leq 0$ and $\frac{\partial L}{\partial x_3} = \frac{\partial L}{\partial x_5}$ otherwise. The derivatives for the parameters $w_{11}$, $w_{12}$, $w_{21}$, and $w_{22}$ function analogously to $w_5$ and $w_6$.

**Parameter Update.** The last step is the parameter update. We also encode the new parameters as a constraint: $w_i' = w_i - \lambda \frac{\partial L}{\partial w_i}$. Theoretically, we could directly continue with the next forward pass, using the new parameters $w_i'$, resulting in an optimization problem that precisely encodes the entire training. However, this is computationally infeasible in practice. We therefore relax the constraints after each parameter update by solving the optimization problem for each parameter: $\underline{w_i'} = \min w_i'$, and $\overline{w_i'} = \max w_i'$, subject to the constraints that encode the forward and backward passes from above. $\underline{w_i'}$ and $\overline{w_i'}$ are real-valued constraints that guarantee $\underline{w_i'} \leq w_i' \leq \overline{w_i'}$. This leads to valid bounds for all parameters in consecutive iterations.

### 3.3 Bounds via Mixed-Integer Bilinear Programming

A key contribution of our method MIBP-Cert is the approach to solving Eq. (4). Using mixed-integer bilinear programming, we can compute an exact solution for each iteration, avoiding over-approximations. For each training iteration, we build an optimization problem over each model parameter, with the new, updated value as the optimization target. $\mathcal{D}$, the current model parameters, the transformation functions of each layer, the loss, the gradients of the backward pass, and the parameter update are encoded as constraints. This results in $2m$ optimization problems—one minimization and one maximization per parameter—for a model with $m$ parameters of the form

$$
\begin{aligned}
\text{min/max} \quad & \theta_{i+1}^j, \quad j = 1, \ldots, m \\
\text{subject to} \quad & \text{Input Constraints} \\
& \text{Parameter Constraints} \\
& \text{Layer Constraints} \\
& \text{Loss Constraints} \\
& \text{Gradient Constraints} \\
& \text{Parameter Update Constraints.}
\end{aligned}
\tag{5}
$$

The objectives are the parameters, which we maximize and minimize independently to compute their upper and lower bounds. The constraints are the same for all parameters and only have to be constructed once. We present these constraints for fully connected models with ReLU activation.

**Input Constraints.** The first set of constraints encodes the allowed perturbations, in this case the $\ell_\infty$-norm with radius $\epsilon$, where $o_k^{(0)}$ are the auxiliary variables encoding the $n$ input features:

$$
o_k^{(0)} \leq x_k + \epsilon, \quad k = 1, \ldots, n, \qquad o_k^{(0)} \geq x_k - \epsilon, \quad k = 1, \ldots, n.
\tag{6}
$$

**Parameter Constraints.** Parameters are bounded from the second iteration, which we encode as:

$$
\theta_i^j \leq \overline{\theta}_i^j, \quad j = 1, \ldots, m \qquad \theta_i^j \geq \underline{\theta}_i^j, \quad j = 1, \ldots, m.
\tag{7}
$$

**Layer Constraints.** *Linear layer* constraints are linear combinations of the layer's inputs $o_v^{(l-1)}$, and the layer's weights $w_{uv}^{(l)} \in \theta_{i-1}$ and biases $b_u^{(l)} \in \theta_{i-1}$:

$$
o_u^{(l)} = \sum_v w_{uv}^{(l)} o_v^{(l-1)} + b_u^{(l)}, \quad u = 1, \ldots, \left| o^{(l)} \right|.
\tag{8}
$$

This results in bilinear constraints, as the layer's parameters are multiplied by the inputs, and both are variables of the optimization problem.

*ReLUs* are encoded as piecewise-linear constraints, e.g., via Big-M [13] or SOS [2]

$$
o_i^{(l)} = \begin{cases} 0 & \text{if } o_u^{(l-1)} \leq 0 \\ o_u^{(l-1)} & \text{otherwise} \end{cases} \quad u = 1, \ldots, \left| o^{(l)} \right|.
\tag{9}
$$

Additional constraints for convolution and other activation functions can be found in Appendix C.

**Loss Constraints.** We use the hinge loss because it is piecewise linear, and we can therefore encode it exactly. For other loss functions, we can use (piecewise) linear relaxations. With the last-layer output $o^{(L)}$, the ground-truth label $y$, and the auxiliary variable $J$, we define the constraint as Eq. (10):

$$
J = \max\left( 0, 1 - yo^{(L)} \right)
\tag{10}
$$

$$
\frac{\partial J}{\partial o^{(L)}} = \begin{cases} -y & \text{if } yo^{(L)} \leq 1 \\ 0 & \text{otherwise} \end{cases}
\tag{11}
$$

**Gradient Constraints.** The gradients of the hinge loss are also piecewise linear (Eq. (11)).

The local gradient of the ReLU function is also piecewise linear (Eq. (12)). Multiplication with the upstream gradient results in a piecewise bilinear constraint (Eq. (13)).

$$
\frac{\partial x_i^{(l)}}{\partial x_i^{(l-1)}} = \begin{cases} 0 & \text{if } x_i^{(l-1)} \leq 0 \\ 1 & \text{otherwise} \end{cases}
\tag{12}
$$

$$
\frac{\partial L}{\partial x_i^{(l-1)}} = \frac{\partial L}{\partial x_i^{(l)}} \frac{\partial x_i^{(l)}}{\partial x_i^{(l-1)}}
\tag{13}
$$

All partial derivatives for linear layers are bilinear:

$$\frac{\partial J}{\partial o_u^{(l-1)}} = \sum_v w_{uv} \frac{\partial L}{\partial o_v^{(l)}} \quad (14a) \qquad \frac{\partial J}{\partial w_{uv}^{(l)}} = o_u^{(l-1)} \frac{\partial J}{\partial o_v^{(l)}} \quad (14b) \qquad \frac{\partial J}{\partial b_u^{(l)}} = o_u \frac{\partial J}{\partial o_u^{(l)}} \quad (14c)$$

**Parameter Update Constraints.** The last set of constraints is the parameter updates. It is essential to include this step before relaxation because the old parameters are contained in both subtraction operands. Solving this precisely is a key advantage compared to prior work (Section 4).

$$\theta_{i+1}^j = \theta_i^j - \lambda \frac{\partial J}{\partial \theta^j}, \quad j = 1, \dots, m. \tag{15}$$

### 3.4 Implementation

We implement MIBP-Cert using Gurobi [14], which provides native support for bilinear and piecewise-linear constraints. To represent parameter bounds throughout training, we use the open-source BoundFlow library [20].

During training, we iteratively build and solve a mixed-integer bilinear program for each model parameter. Each program encodes the input perturbation model, forward pass, loss, backward pass, and parameter update step, capturing the complete symbolic structure of one training iteration. Crucially, unlike interval methods, we preserve operand dependencies throughout, allowing bounds to tighten dynamically.

After training, we use the final parameter bounds to certify predictions. At inference time, we encode only the forward pass with symbolic parameters bounded by $[\underline{\theta}, \overline{\theta}]$. If one logit is provably larger than all others across this range, we return its class label; otherwise, the model abstains.

A full pseudocode listing is provided in Appendix B, and additional implementation details can be found in Appendix D.

## 4 Bound Convergence Analysis

Prior certified training methods based on interval or polyhedral relaxations [20, 28] suffer from significant over-approximations introduced after each layer. While these approximations are effective for inference-time certification [4, 12, 27], they require robust training to compensate [22].

In training-time certification, the situation is different. Over-approximations accumulate across iterations and cannot be corrected during training. Lorenz et al. [20] show that these accumulated errors can prevent convergence, even at a local minimum, resulting in exploding parameter bounds.

This behavior is analyzed using the Lyapunov sequence $h_i = \|\theta_i - \theta^*\|^2$, which measures the distance to the optimum. For standard SGD [6], the recurrence is

$$h_{i+1} - h_i = \underbrace{-2\lambda_i(\theta_i - \theta^*)\nabla_\theta J(\theta_i)}_{\text{distance to optimum}} + \underbrace{\lambda_i^2 (\nabla_\theta J(\theta_i))^2}_{\text{discrete dynamics}}. \tag{16}$$

Under common assumptions (bounded gradients, decaying learning rate), the second term is bounded. The key requirement is that the first term remains negative to guarantee convergence.

Lorenz et al. [20] show that this condition may fail under interval arithmetic: if the current parameter bound $\Theta_i$ and the (unknown) optimum $\Theta^*$ overlap, the worst-case inner product in Eq. (16) may vanish or become positive.

MIBP-Cert avoids this issue by maintaining the exact symbolic structure of the update step. Rather than reasoning over interval bounds, we evaluate the convergence condition for all possible realizations $\theta_i \in \Theta_i, \theta^* \in \Theta^*$. As shown by Bottou [6], the convergence term remains negative for every such pair under the convexity assumption, ensuring that training remains stable.

A second limitation noted by Lorenz et al. [20] is that parameter intervals can only grow when both operands of the addition are intervals: $\theta_{i+1} = \theta_i - \lambda \nabla_\theta J(\theta_i) \Rightarrow |\theta_{i+1}| = |\theta_i| + |\lambda \nabla_\theta J(\theta_i)|$. This leads to expanding bounds over time, regardless of convergence.

MIBP-Cert overcomes this by computing exact parameter updates (Eq. 15), preserving dependencies and allowing parameter bounds to shrink. As a result, we do not observe divergence or instability in

practice. While the exact formulation is more computationally expensive, it leads to significantly tighter bounds and more stable certified training.

## 5 Experiments

We evaluate MIBP-Cert on certified accuracy, runtime, and support for expressive threat models, comparing it to prior methods across multiple datasets.

### 5.1 $\ell_\infty$-Perturbations of Continuous Features

We evaluate MIBP-Cert experimentally and compare it to the state of the art in deterministic training certification: FullCert [20] and Sosnin et al. [28].

**TwoMoons.** We evaluate on the Two-Moons dataset—two classes configured in interleaving half circles—as it is used in prior work. Table 2 presents the certified accuracy, i.e., the percentage of data points from a held-out test set where Algorithm 2 returns the ground-truth class. All values are the mean and standard deviation across different iid seeds.

For small perturbation radii $\epsilon$, our method performs similarly to the baselines. With increasing radius, the advantages of tighter bounds become apparent, where our method significantly outperforms the baselines. This trend makes sense, as the over-approximations become more influential with larger perturbations. The second advantage of our method becomes apparent when looking at the standard deviations. The small standard deviation compared to the baselines shows a much more stable training behavior, which aligns with our analysis in Section 4.

| $\epsilon$ | 0.0001 | 0.001 | 0.01 |
|---|---|---|---|
| Sosnin et al. | 83.8%±0.02 | 82.3%±0.03 | 69.0%±0.10 |
| FullCert | 83.9%±3.60 | 82.2%±4.40 | 71.5%±11.20 |
| MIBP-Cert (ours) | 83.3%±0.05 | 82.0%±0.05 | 81.4%±0.06 |

Table 2: Comparison of MIBP-Cert to FullCert [20] and Sosnin et al. [28] for different $\epsilon$ values. The numbers represent the mean and standard deviation of certified accuracy across random seeds.

**UCI Datasets.** To show the transferability of these results to different datasets, we evaluate our method on two additional datasets in the same threat model: the Iris [11] and Breast Cancer Wisconsin [35]. Table 3a shows certified accuracy for a 2-class subset of the Iris dataset. Even for large perturbation radii of 0.1 after standardization, we guarantee correct prediction for all test points. The radius decreases for larger perturbation radii, dropping below 50% for $\epsilon = 0.3$. Table 3b shows the generalization of our method to multi-class classification. Table 3c (right) demonstrates the certified accuracy for breast cancer diagnostics. MIBP-Cert shows that the larger feature count gives the adversary more control, which decreases the certified accuracy for larger perturbations.

| $\epsilon$ | 0.1 | 0.2 | 0.3 |
|---|---|---|---|
| FullCert | 88% | 76% | 20% |
| MIBP-Cert | 100% | 92% | 40% |

| $\epsilon$ | 0.00 | 0.01 | 0.02 |
|---|---|---|---|
| FullCert | 84% | 68% | 64% |
| MIBP-Cert | 84% | 72% | 68% |

| $\epsilon$ | 0.000 | 0.001 | 0.010 |
|---|---|---|---|
| FullCert | 95% | 83% | 38% |
| MIBP-Cert | 95% | 94% | 51% |

    (a) Iris 2-class                 (b) Iris full                 (c) Breast Cancer

Table 3: Certified accuracy of MIBP-Cert (ours) and FullCert [20] for training a 2-layer MLP on different UCI datasets [16].

### 5.2 Complex Constrained Perturbations on Discrete Features

We investigate how to understand and provide robustness when training on health data that has been shown to be prone to corruptions, such as missing data and biases in self-reporting [1]. Unlike prior work, MIBP-Cert's general formulation supports fine-grained perturbation models beyond simple $\ell_p$-norms. We demonstrate this on the *National Poll on Healthy Aging (NPHA)* [21], a tabular dataset of 714 survey responses from adults over 50. It includes 14 self-reported categorical features covering mental and physical health, sleep quality, and employment. We evaluate three structured

perturbation scenarios that reflect realistic sources of data uncertainty and use MIBP-Cert to compute the worst-case impact of each. Table 4 reports the *certification rate* (fraction of test samples with a verified prediction) and *certified accuracy* (fraction of certified samples with the correct label).

| Precondition | Certification Rate | Certified Accuracy |
|---|---|---|
| Assuming accurate health data | (100.0%) | 56.3% |
| Modeling missing mental health values | 98.6% | 56.3% |
| Modeling missing values across all features | 95.8% | 53.5% |
| Modeling mental health over-reporting | 91.5% | 50.7% |

Table 4: Certification under complex perturbation models. We vary the allowed perturbations (preconditions) to assess their impact on prediction robustness.

**Missing mental health values.** Some participants declined to answer questions on mental health, possibly introducing bias (e.g., lower-scoring individuals may be more likely to abstain). We model this by replacing the "no answer" option with a perturbation model of any valid response: $x[j] \in \{0, 1\}, \sum_j x_j = 1$. Despite this uncertainty, 98.6% of the predictions are certifiable (guaranteed not to be affected by the missing values in the training data), and the certified accuracy is unaffected.

**Missing values across all features.** Extending this idea, we model all missing values across the 14 features as perturbations that can take any valid value. The impact is larger, reducing certification to 95.8% and certified accuracy to 53.5%, indicating that imputation uncertainty can meaningfully affect model behavior—but on the other hand we still know the (certified) test samples for which the prediction cannot change.

**Mental health over-reporting.** Self-assessments are prone to bias [1]. We simulate optimistic self-reporting by allowing mental health values above the smallest value to be either the reported value $k$ or the next worse value $k-1$: $\forall k = \{2, \ldots, 5\} : x[j] = 0 \ \forall j \notin \{k, k-1\}, \ x[k], x[k-1] \in \{0, 1\}, \ x[k] + x[k-1] = 1$. The result: 91.5% of the predictions remain certifiable, but in 8.5% of the cases, optimistic self-assessment in the training data can alter the model's prediction. This highlights the importance of accounting for reporting bias in downstream conclusions.

## 5.3 Runtime and Complexity Analysis

MIBP-Cert achieves tighter bounds than polytope-based methods, at the cost of increased optimization complexity. Solving a mixed-integer bilinear program (MIBP) is NP-hard in general [3, 17].

In our formulation, binary variables arise from ReLU activations and $\max$ terms in the loss, yielding approximately $(n+k)b$ binary decisions per iteration, where $n$ is the number of activations, $k$ is the number of classes, and $b$ is the batch size. Each optimization subproblem thus combines combinatorial branching over these binary variables with continuous bilinear relaxations that depend on the number of model parameters $p$. In theory, the resulting search tree thus grows exponentially in $(n+k)b$ and superlinearly with $p$. But modern solvers such as Gurobi employ presolve, bound tightening, and outer-approximation heuristics that drastically prune this tree in practice, making solutions feasible for most problems.

| $\epsilon$ | 0.01 | 0.1 |
|---|---|---|
| batch-size 10 | 4 | 36 |
| batch-size 100 | 116 | 174552 |

Table 5: The number of branches explored by the optimizer during the first two training iterations, for different batch sizes and perturbation radii. Although theoretical limits are exponential, empirical counts remain tractable.

Empirical runtimes remain several orders of magnitude below the theoretical worst case, even for large batch sizes or perturbation radii (Table 5). Perturbation radius $\epsilon$ primarily affects complexity through the number of active ReLU switches, and early epochs are typically faster due to tighter parameter bounds. A full runtime breakdown is provided in Appendix E.2.

# 6 Discussion and Limitations

MIBP-Cert addresses key limitations of prior certified training methods by avoiding coarse convex over-approximations that can lead to diverging bounds and unstable learning. Using mixed-integer bilinear programming (MIBP), it computes tight bounds at each iteration, ensuring that parameter intervals can shrink over time and remain stable.

The primary benefit of this increased tightness is improved certified accuracy, particularly under larger perturbation radii. Our experiments show that MIBP-Cert not only yields higher certified accuracy but also exhibits lower variance, indicating more robust behavior across runs.

Beyond accuracy, the flexibility of our MIBP formulation enables more expressive threat models. As demonstrated in Section 5.2, it can encode structured constraints and domain-specific assumptions, which was impossible with prior methods. This opens the door to modeling realistic attacks and non-adversarial perturbations alike, such as measurement noise or reporting bias in tabular domains.

These tighter, more expressive bounds come at the cost of a higher complexity, and, thus, computational cost: solving a bilinear program per parameter per step is more expensive than layer-wise interval or polytope propagation. We demonstrate that our method can be applied to a range of safety, security, or privacy-critical problems used in practice, and we expect that follow-up work can improve on the scalability of our prototype implementation, similarly to how test-time certification methods have improved substantially over early, expensive methods that were limited to small model sizes as well. Potentially impactful directions include: hybrid methods combining our tight bounds selectively with faster approximations elsewhere, exploiting model sparsity and structural properties for faster optimization, exploring low-level engineering efficiency gains through optimized and/or parallelized implementations, and improvements in how to guide optimizer heuristics by exploiting domain/problem knowledge.

Overall, our results suggest that high-precision certified training is not only feasible but necessary to move beyond current limitations—particularly when robustness must be guaranteed under complex perturbation models.

# 7 Conclusion

We present MIBP-Cert, a certified training approach that overcomes the limitations of prior work by avoiding loose convex over-approximations and enabling exact parameter bounds via MIBP. This leads to significantly improved certified accuracy and training stability, especially under larger perturbations and expressive threat models. Our results demonstrate that precise, structured certification is not only feasible but essential for robust learning under real-world data corruptions and attacks.

## Acknowledgments and Disclosure of Funding

We thank the NeurIPS reviewers and area chairs for their valuable feedback. This work was partially funded by ELSA—European Lighthouse on Secure and Safe AI funded by the European Union under grant agreement number 101070617, as well as the German Federal Ministry of Education and Research (BMBF) under the grant AIgenCY (16KIS2012), and Medizininformatik-Plattform "Privatsphären-schützende Analytik in der Medizin" (PrivateAIM), grant number 01ZZ2316G. MK received funding from the ERC under the European Union's Horizon 2020 research and innovation programme (FUN2MODEL, grant agreement number 834115).

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

## A  Full Example

In Section 3.2 we present an example model to illustrate our method. We list the full optimization problem for the first training iteration here.

$$\text{min/max} \quad w_i', \quad i \in \{11, 12, 21, 22, 5, 6\}$$

subject to

$$0 \le x_1 \le 2$$
$$0 \le x_2 \le 2$$
$$1 \le w_{11} \le 1$$
$$1 \le w_{12} \le 1$$
$$1 \le w_{21} \le 1$$
$$-1 \le w_{22} \le -1$$
$$-1 \le w_5 \le -1$$
$$1 \le w_6 \le 1$$
$$x_3 = w_{11}x_1 + w_{21}x_2$$
$$x_4 = w_{12}x_1 + w_{22}x_2$$
$$x_5 = \begin{cases} x_3 & \text{if } x_3 > 0 \\ 0 & \text{otherwise} \end{cases}$$
$$x_6 = \begin{cases} x_4 & \text{if } x_4 > 0 \\ 0 & \text{otherwise} \end{cases}$$
$$x_7 = w_5 x_5 + w_6 x_6$$
$$L = \begin{cases} 1 - x_7 & \text{if } 1 - x_7 > 0 \\ 0 & \text{otherwise} \end{cases}$$
$$\frac{\partial L}{\partial x_7} = \begin{cases} -1 & \text{if } 1 - x_7 > 0 \\ 0 & \text{otherwise} \end{cases}$$
$$\frac{\partial L}{\partial w_6} = x_6 \frac{\partial L}{\partial x_7}$$
$$\frac{\partial L}{\partial x_6} = w_6 \frac{\partial L}{\partial x_7}$$

$$\frac{\partial L}{\partial w_5} = x_5 \frac{\partial L}{\partial x_7}$$
$$\frac{\partial L}{\partial x_5} = w_5 \frac{\partial L}{\partial x_7}$$
$$\frac{\partial L}{\partial x_4} = \begin{cases} \frac{\partial L}{\partial x_6} & \text{if } x_4 > 0 \\ 0 & \text{if otherwise} \end{cases}$$
$$\frac{\partial L}{\partial x_3} = \begin{cases} \frac{\partial L}{\partial x_5} & \text{if } x_3 > 0 \\ 0 & \text{if otherwise} \end{cases}$$
$$\frac{\partial L}{\partial w_{11}} = x_1 \frac{\partial L}{\partial x_3}$$
$$\frac{\partial L}{\partial w_{12}} = x_1 \frac{\partial L}{\partial x_4}$$
$$\frac{\partial L}{\partial w_{21}} = x_2 \frac{\partial L}{\partial x_3}$$
$$\frac{\partial L}{\partial w_{22}} = x_2 \frac{\partial L}{\partial x_4}$$
$$\frac{\partial L}{\partial x_2} = w_{21} \frac{\partial L}{\partial x_3} + w_{22} \frac{\partial L}{\partial x_4}$$
$$\frac{\partial L}{\partial x_1} = w_{11} \frac{\partial L}{\partial x_3} + w_{12} \frac{\partial L}{\partial x_4}$$
$$w_i' = w_i - \lambda \frac{\partial L}{\partial w_i}$$

## B  Training and Prediction Algorithms

We implement the optimization procedure outlined in Section 3.4 according to Algorithm 1. For each iteration of the training algorithm, we initialize an optimization problem (5) and add the parameter bounds as constraints (6). For each data point $x$, we add the input constraints (8), layer constraints (10), loss constraints (12), gradient constraints (13-15), and parameter update constraints (18). We then solve the optimization problem for each parameter, once for the upper and once for the lower bound (19-20). The algorithm returns the final parameter bounds.

Once the model is trained, we can use the final parameter bounds for prediction (Algorithm 2). The principle is the same as encoding the forward pass for training (3-7). For classification, we can then compare the logit and check whether one is always greater than all others (9-14). If so, we return the corresponding class (16). Otherwise, we cannot guarantee a prediction and return *abstain* (19).

## C  Additional Constraints

Our method for bounding neural network functions introduced in Section 3.1 is general and can be extended to layer types beyond linear and ReLUs introduced in Section 3.3, as we show on the examples of convolution and general activation functions:

**Algorithm 1** MIBP Train
---
1: **Input:** dataset $D$, initial parameters $\theta_0 \in \mathbb{R}^m$, iterations $n \in \mathbb{N}$
2: **Output:** parameter bounds $\underline{\theta}_n, \overline{\theta}_n \in \mathbb{R}^m$
3: Initialize $\underline{\theta}_0, \overline{\theta}_0 \leftarrow \theta_0$
4: **for** $i = 1$ **to** $n$ **do**
5:     mibp $\leftarrow$ *initialize_optimization*()
6:     mibp.*add_parameter_constraints*($\underline{\theta}_{i-1}, \overline{\theta}_{i-1}$)
7:     **for** $x \in D$ **do**
8:        mibp.*add_input_constraints*($x$)
9:        **for** each layer $l = 1$ **to** $L$ **do**
10:          mibp.*add_layer_constraints*($l$)
11:        **end for**
12:        mibp.*add_loss_constraints*()
13:        mibp.*add_loss_gradient_constraints*()
14:        **for** each layer $l = L$ **to** $1$ **do**
15:          mibp.*add_gradient_constraints*($l$)
16:        **end for**
17:     **end for**
18:     mibp.*add_parameter_update_constraints*()
19:     $\underline{\theta}_i = $ mibp.*minimize*($\theta_i$)
20:     $\overline{\theta}_i = $ mibp.*maximize*($\theta_i$)
21: **end for**
22: **Return** $\underline{\theta}_n, \overline{\theta}_n$

**Algorithm 2** MIBP Predict
---
1: **Input:** test data $x$, parameter bounds $\underline{\theta}, \overline{\theta} \in \mathbb{R}^m$
2: **Output:** certified prediction $y$, or *abstain*
3: mibp $\leftarrow$ *initialize_optimization_problem*()
4: mibp.*add_parameter_constraints*($\underline{\theta}, \overline{\theta}$)
5: mibp.*add_input_variables*($x$)
6: **for** layer $l = 1$ **to** $L$ **do**
7:     mibp.*add_layer_constraints*($l$)
8: **end for**
9: **for each** logit $o_u^{(L)}$ **do**
10:     $c \leftarrow$ True
11:     **for each** logit $o_v^{(L)} \neq o_u^{(L)}$ **do**
12:        $c_v = $ mibp.*minimize*($o_u^{(L)} - o_v^{(L)}$)
13:        $c \leftarrow c \wedge (c_v \geq 0)$
14:     **end for**
15:     **if** $c$ **then**
16:        **Return** $u$
17:     **end if**
18: **end for**
19: **Return** *abstain*

**Convolution.** Convolution layers are linear operations and can be encoded as linear constraints. Given an input tensor $X \in \mathbb{R}^{C_{\text{in}} \times H \times W}$ and a kernel $K \in \mathbb{R}^{C_{\text{out}} \times C_{\text{in}} \times k_H \times k_W}$, the convolution output $Y \in \mathbb{R}^{C_{\text{out}} \times H' \times W'}$ is defined by:

$$Y_{c,i,j} = \sum_{c'=1}^{C_{\text{in}}} \sum_{m=1}^{k_H} \sum_{n=1}^{k_W} K_{c,c',m,n} \cdot X_{c',\,i+m-1,\,j+n-1} + b_c. \qquad (17)$$

This operation is equivalent to a sparse affine transformation and can be flattened into a set of linear constraints $Y = AX + b$, where $A$ encodes the convolution as a sparse matrix multiplication and $b$ is the bias. These constraints can be directly integrated into our MIBP formulation.

**General Activations.**  MIBP-Cert also supports other non-linear activation functions (e.g., sigmoid, tanh) by employing piecewise-linear upper and lower bounds over the relevant input range. For example, in the range $[0, 1]$:

- Sigmoid: $0.25x + 0.48 \leq \sigma(x) \leq 0.25x + 0.5$
- Tanh: $0.76x \leq \tanh x \leq x$.

Although these relaxations introduce some degree of over-approximation, their piecewise-linear formulation allows the bounds to be made arbitrarily tight.

## D  Training and Implementation Details

**Implementation Details.**  We build on Lorenz et al. [20]'s open-source library BoundFlow with an MIT license, which integrates with PyTorch [24] for its basic tensor representations and arithmetic. As a solver backend, we use Gurobi version 10.0.1 with an academic license. Gurobi is a state-of-the-art commercial solver that is stable and natively supports solving mixed-integer bilinear programs. It also provides a Python interface for easy integration with deep learning pipelines.

**Compute Cluster.**  All computations are performed on a compute cluster, which mainly consists of AMD Rome 7742 CPUs with 128 cores and 2.25 GHz. Each task is allocated up to 32 cores. No GPUs are used since Gurobi does not use them for solving.

**Dataset Details.**

*Two Moons (Synthetic)* We use the popular Two Moons dataset, generated via scikit-learn [25]. It is a 2D synthetic binary classification task with non-linear decision boundaries, commonly used to visualize model behavior and certification properties. We set the noise parameter to $0.1$ and sample 100 points for training, 200 points for validation, and 200 points for testing, respectively.

*UCI Iris [11]* The Iris dataset is a classic multi-class classification benchmark with 150 samples and 4 continuous features. We experiment with both the full 3-class setting (100 train, 25 validation, 25 test) and a reduced binary subset of the first two classes (using 50 train, 25 validation, 25 test). The low input dimensionality and categorical nature of the target make it a useful testbed for analyzing certified training on tabular data.

*UCI Breast Cancer Wisconsin [35]* We use the UCI Breast Cancer Wisconsin dataset (binary classification) with 30 continuous input features. We split the data into 369 training, 100 validation, and 100 test samples. The dataset allows us to study our method's scalability to medium-sized tabular data and its performance under realistic feature distributions and class imbalances.

The *National Poll on Healthy Aging (NPHA)* [21] is a tabular medical dataset with 14 categorical features and 3 target classes. The features represent age, physical health, mental health, dental health, employment, whether stress, medication, pain, bathroom needs, or unknown reasons keep patients from sleeping, general trouble with sleeping, the frequency of taking prescription sleep medication, race, and gender. The target is the frequency of doctor visits. We randomly (iid) split the data points into 3 independent sets, with 10 % for validation, 10 % for testing, and the remainder for training.

**Model Architecture.**  Unless indicated otherwise, we use fully connected networks with ReLU activations, two layers, and 20 neurons per layer. For binary classification problems, we use hinge loss, i.e., $J = \max(0, 1 - y \cdot f(x))$, because it is piecewise linear and can therefore be encoded exactly. It produces similar results to Binary Cross-Entropy loss for regular training without perturbations.

**Training Details.**  We train models until convergence using a held-out validation set, typically after 5 to 10 epochs on Two-Moons. We use a default batch size of 100 and a constant learning rate of 0.1. We sub-sample the training set with 100 points per iteration. All reported numbers are computed using a held-out test set that was not used during training or hyper-parameter configuration.

**Comparison with Prior Work.**  For the comparison to Lorenz et al. [20], we use the numbers from their paper. Since Sosnin et al. [28] do not report certified accuracy for Two-Moons, we train new models in an equivalent setup. As a starting point, we use the Two-Moons configurations provided in their code. We change the model architecture to match ours, i.e., reducing the number of hidden neurons to 20. We also set $n = |D|$ to adjust the threat model to be the same as ours. The solver mode is left at its preconfigured "interval+crown" for the forward pass and "interval" for the backward pass.

Contrary to Sosnin et al. [28], we do not increase the separation between the two classes compared to the default scikit-learn implementation. We also do not add additional polynomial features.

## E  Additional Experiments

### E.1  Model Architecture Ablations

Section 5.3 shows that the complexity of MIBP-Cert mainly depends on the number of activations $n$, the number of classes $k$, and the batch size $b$. Table 5 demonstrates the influence of batch size and perturbation size experimentally. Here, we empirically analyze the influence of the model width and depth in Tables 6 and 7, which both influence the number of parameters and activations. We use the same models and data as in Table 2, with $\epsilon = 0.01$, training for 1 epoch with varying network width and depth.

| Layers $\times$ width | Total parameters | Training time |
|---|---|---|
| $2 \times 5$ | 27 | 0.75 s |
| $2 \times 10$ | 52 | 1.24 s |
| $2 \times 20$ | 102 | 4.79 s |
| $2 \times 30$ | 152 | 27.7 s |

Table 6: MIBP-Cert's runtime for 1 epoch with different layer widths.

| Layers $\times$ width | Total parameters | Training time |
|---|---|---|
| $2 \times 5$ | 27 | 0.75 s |
| $3 \times 5$ | 57 | 2.37 s |
| $4 \times 5$ | 87 | 6.24 s |
| $5 \times 5$ | 117 | 12.9 s |
| $6 \times 5$ | 147 | 22.1 s |

Table 7: MIBP-Cert's runtime for 1 epoch with different numbers of layers.

The results show that training time corresponds roughly to the number of parameters.

### E.2  Runtime Comparison

To compare the real-world execution time of MIBP-Cert with that of prior work, we report the average runtime per epoch on the Two-Moons dataset, in the same setting as shown in Table 2. FullCert and Sosnin et al. in pure IBP mode both take 0.014s per epoch. IBP+CROWN mode (the default for TwoMoons) takes 0.04s per epoch, and CROWN takes 0.05s per epoch. This is relatively consistent across different perturbation radii ($\epsilon$) and epochs. We report the average runtime for the first 5 epochs in Table 8. Training runs typically either converge or diverge after 5 epochs.

| Method | $\epsilon = 0.0001$ | | $\epsilon = 0.001$ | | $\epsilon = 0.01$ | |
|---|---|---|---|---|---|---|
| | Time | CA | Time | CA | Time | CA |
| FullCert | 0.01 s | 83.9% | 0.01 s | 82.2% | 0.01 s | 71.5% |
| CROWN+IBP | 0.04 s | 83.8% | 0.04 s | 82.3% | 0.03 s | 69.0% |
| MIBP-Cert (ours) | 19 s | 83.3% | 54 s | 82.0% | 156 s | 81.4% |

Table 8: Runtime of different certified training methods for a 2-layer MLP on TwoMoons. "Time" is the average runtime per epoch across the first 5 epochs, and "CA" is the certified accuracy.

While our method is slower, especially for larger $\epsilon$, it delivers significantly improved certified accuracy and training stability, validating our theoretical analysis regarding the runtime-precision trade-off. Furthermore, as Table 1 and Section 5.2 demonstrate, our method uniquely certifies perturbation types that were previously impossible.

