# OpenReview forum: "MIBP-Cert: Certified Training against Data Perturbations with Mixed-Integer Bilinear Programs"
_NeurIPS.cc/2025/Conference — NeurIPS 2025 poster_

### Official Review · Reviewer_VCxT · 2025-06-11

**Clarity:** 3
**Significance:** 2
**Originality:** 2
**Rating:** 3
**Confidence:** 3

**Summary:**

This work proposes a certified training method against data perturbations via mixed integer bilinear (MIBP) programs. At each training step, the authors formulate the gradient descent process as an MIBP problem and solve it using Gurobi. It allows them to bound the range of the updated model parameters the when the training examples are perturbed. By iteratively applying this process for all steps, the authors can obtain the range of the final model parameters, which is served as a certification.

**Questions:**

1. What is the running time for your method and previous methods?
2. How does the running time of your method scale as the model size increases?

**Ethical Concerns:**

["NO or VERY MINOR ethics concerns only"]

**Final Justification:**

The authors have addressed my concern. However, given the time complexity of the proposed approach, I have decided to maintain my current score.

**Limitations:**

Yes.

**Quality:**

2

**Strengths And Weaknesses:**

Strengths
1. This paper is well-written and easy to follow. The proposed example is clear and helpful.
2. The proposed MIBP-Cert method shows tighter bound than previous methods, particularly under large perturbations.

Weaknesses
1. The proposed method exhibits higher time complexity than previous approaches. Moreover, as the neural network scales, the number of constraints in the MIBP formulation grows, making it impractical for more complex architectures.
2. There is no running time comparison with previous methods.
3. The model is a 2-layer MLP, which is too simple.

---

> ### Author Rebuttal · Authors · 2025-07-31
>
> Dear Reviewer VCxT,
>
> Thank you for taking the time to review our paper! We are encouraged that you found it easy to follow and appreciate the more precise bounds of our approach. We address your questions and concerts below.
>
> &nbsp;
> ### 1. What is the running time for your method and previous methods?
>
> Thank you for this suggestion, we agree that a runtime comparison makes sense to illustrate the trade-offs involved. To address this, **we report the average runtime per epoch on the Two-Moons dataset**, in the same setting as Table 1. The detailed runtime of our method across epochs and perturbation radii is provided in Appendix C of our manuscript. FullCert and Sosnin et al. in pure IBP mode both take ~0.014s per epoch. IBP+CROWN mode (the default for TwoMoons) takes ~0.04s per epoch, and ~0.05s per epoch for CROWN. This is relatively consistent across different perturbation radii ($\epsilon$) and epochs. We report the average runtime for the first 5 epochs in the table below. Models typically either converge or diverge after 5 epochs.
>
> | Method | Time ($\epsilon=0.0001$) | CA ($\epsilon=0.0001$) | Time ($\epsilon=0.001$) | CA ($\epsilon=0.001$) | Time ($\epsilon=0.01$) | CA ($\epsilon=0.01$) |
> |---|---|---|---|---|---|---|
> | FullCert | 0.01s |83.9%|0.01s |82.2%|0.01s |71.5%|
> | Sosnin et al.| 0.04s |83.8%|0.04s |82.3%|0.03s |69.0%|
> | MIBP-Cert | 19s |83.3%|54s |82.0%|156s |81.4%|
>
> While our method is slower, especially for larger $\epsilon$, it delivers significantly improved certified accuracy and training stability, validating our theoretical analysis regarding the runtime-precision trade-off. Furthermore, as Table 1 and Section 5.2 demonstrate, our method **uniquely certifies perturbation types that were previously impossible**. We will include an expanded runtime comparison in Section 5.3.
>
> &nbsp;
> ### 2. How does the running time scale as the model size increases?
>
> Our complexity analysis in section 5.3 shows that the complexity bound is $O(2^{(n + k)b} \cdot 2p)$, meaning **the dominant factors are the number of activations $n$ and the number of classes $k$**. Therefore, scalability depends on these two values.
>
> To analyze real-world behaviour we run ablation studies on the model width and depth, which both influence the number of parameters and activations. We use the same models and data as in Table 1 with $\epsilon=0.01$, training for 1 epoch with varying network width and depth.
>
> | layers x width | total parameters | Training time |
> |---|---|---|
> | 2 x 5 | 27 | 0.75s |
> | 2 x 10 | 52 | 1.24s |
> | 2 x 20 | 102 | 4.79s |
> | 2 x 30 | 152 | 27.7s |
>
> | layers x width | total parameters | Training time |
> |---|---|---|
> | 2 x 5 | 27 | 0.75s |
> | 3 x 5 | 57 | 2.37s |
> | 4 x 5 | 87 | 6.24s |
> | 5 x 5 | 117 | 12.9s |
> | 6 x 5 | 147 | 22.1s |
>
> The numbers show that **training time corresponds roughly to the number of parameters**.
>
> &nbsp;
> ### 3. The model is a 2-layer MLP, which is too simple.
>
> We agree that **larger models would indeed be desirable**. Nevertheless, we demonstrate that smaller architectures are being **applied in safety-, security-, or privacy-critical scenarios**, where verifiable guarantees are crucial and worth the trade-offs. We expect that follow-up work can improve on scalability by building on our prototype solution to develop tailored algorithmic schemas that are amenable to parallelization and optimization, similarly to how test-time certification methods have improved substantially over early, expensive methods that were limited to small model sizes as well. For example, **our results on the suggestion by reviewer 4X7Z to restrict certified training to only the classification head look promising** towards scaling to more complex models.
>
> &nbsp;
> ### 4. Higher time complexity compared to previous approaches.
>
> Yes. The more precise bounds of our method are an intentional design decision of our method, which **mitigates convergence issues** inherent in prior work, as shown in Section 4. Since **there is no free lunch in optimization** [a], this necessitates a higher theoretical complexity bound. However, as mentioned before, we expect that follow-up work can improve the scalability of our prototype implementation to develop tailored algorithmic schemas that are amenable to parallelization and optimization. Potentially impactful directions include:
> - Hybrid methods combining our tight bounds selectively with faster approximations elsewhere
> - Exploiting model sparsity and structural properties for faster optimization
> - Exploring low-level engineering efficiency gains through optimized implementations
> - Improvements in how to guide optimizer heuristics by exploiting domain/problem knowledge
>
> We will include this discussion in the revision.
>
> &nbsp;
>
> Thank you again for your valuable feedback, which helps to significantly strengthen the revision of our paper.

---

> > ### Comment · Reviewer_VCxT · 2025-08-05
> >
> > Thanks for your reply. It addresses my concerns. However, I decide to maintain my score due to the high computational complexity.

---

> ### Author Response · Authors · 2025-08-06
>
> Dear Reviewer VCxT,
>
> Thank you for your response to our rebuttal, we are glad that it addresses your concerns.
>
> While we take the scalability challenge seriously, we respectfully argue that appropriate weight should be given to the problem's importance, as well as the method’s clear novelty, correctness, and demonstrated advantages over prior work.
>
> We believe that foundational methods such as ours—which for the first time enable certified training guarantees with stable convergence and for previously uncertifiable perturbation types—represent a significant advancement in the science of robustness. Such contributions should be judged in the context of works in this area, where many impactful certification methods initially started with small models and high runtimes, and only later scaled through sustained follow-up work [a, b, c]. We view our work as a critical first step that opens up new directions for certifiably robust training.
>
> Our current solution's high computational complexity is a known limitation, but it is clearly acknowledged, discussed, and accompanied by concrete suggestions for improvement. We are confident that our formulation and findings contribute meaningfully to the scientific foundation on which more scalable and practical approaches can be built.
>
> Thank you for your consideration—we would be happy to discuss this further if needed.
>
> &nbsp;
>
> [a] Katz, Guy, et al. "Reluplex: An efficient SMT solver for verifying deep neural networks." International conference on computer aided verification. Cham: Springer International Publishing, 2017.
>
> [b] Tjeng, Vincent, Kai Y. Xiao, and Russ Tedrake. "Evaluating Robustness of Neural Networks with Mixed Integer Programming." International Conference on Learning Representations.
>
> [c] Brix, Christopher, et al. "The fifth international verification of neural networks competition (vnn-comp 2024): Summary and results." arXiv preprint arXiv:2412.19985 (2024).

---

### Official Review · Reviewer_mYxi · 2025-06-30

**Clarity:** 3
**Significance:** 2
**Originality:** 1
**Rating:** 2
**Confidence:** 3

**Summary:**

The paper introduces a mixed-integer bilinear programming formulation to train ReLU networks with provable robustness. The idea is to compute a reachable set of network parameters by propagating constraints through the network. The experimental results show a slight improvement on toy networks and datasets.

**Questions:**

In general, I would suggest to evaluate possible adaptation of branch & bound to the actual problem formulation to speed up the
- Would it be possible to extend the discussion to other neural network architectures (convolutions, normalization, etc..)?
- Would it be possible to extend the evaluation to MNIST & CIFAR10 datasets.
- Please report also timing of FullCert and MIBP-Cert

**Ethical Concerns:**

["NO or VERY MINOR ethics concerns only"]

**Final Justification:**

While the proposed method shows some promise in robustness certification, the improvements are too incremental to justify publication. In my opinion, the gains do not sufficiently advance the field.

**Limitations:**

The limitations have been discussed in section 6.

**Paper Formatting Concerns:**

There are no concerns

**Quality:**

3

**Strengths And Weaknesses:**

### Strengths
- The idea is clear and properly explained
- The formulation is comprehensive and produces tight bounds
- The problem is relevant and important for safety critical applications

### Weaknesses
- The mixed-integer bilinear optimization formulation has only been investigated for ReLU fully connected networks.
- Scalability is a major concern. Despite optimized bound propagation, the resulting formulation becomes intractable even for networks of modest size. Timing have not been reported
- Experimental results have been only evaluated on toy data.

---

> ### Author Rebuttal · Authors · 2025-07-31
>
> Dear Reviewer mYxi,
>
> Thank you for taking the time to review our paper! We are glad that you found it relevant, comprehensive, and clearly explained. We address your questions and concerns below.
>
> &nbsp;
> ### 1. Adaptation of branch & bound algorithms
>
> Thank you for the suggestion! We agree that the off-the-shelf optimizer, while generally efficient, can potentially be optimized by developing tailored algorithmic schemas that exploit the specific program structure. **The goal of this work is to show that stable certified training is possible** using our novel Mixed Integer Bilinear Programming formulation. Our prototype implementation demonstrates that it can be applied to a range of safety-, security-, or privacy-critical problems. We expect that follow-up work can improve on the scalability of our implementation, similarly to how test-time certification methods have improved substantially over early, expensive methods that were limited to small model sizes as well. Potentially impactful directions include:
>
> - Exploiting model sparsity and structural properties for faster optimization
> - Hybrid methods combining our tight bounds selectively with faster approximations elsewhere
> - Exploring low-level engineering efficiency gains through optimized and/or parallelized implementations
> - Improvements in how to guide optimizer heuristics by exploiting domain/problem knowledge
>
> We will include this discussion in the revision.
>
> &nbsp;
> ### 2. Can you extend the discussion to other network architectures?
>
> Yes. We follow prior work on certification and focus on fully connected models with ReLU MLPs, as this is a very general architecture and well-suited for certification. **Solving the problem for ReLU is therefore the first step to piecewise linear bounding of general activation functions**. However, **our method is more general and can support other architectures, which we are happy to include and discuss in the revision.**
>
> **Convolution** layers are linear operations and can be encoded as linear constraints. Given an input tensor $X \in \mathbb{R}^{C_{\text{in}} \times H \times W}$ and a kernel $K \in \mathbb{R}^{C_{\text{out}} \times C_{\text{in}} \times k_H \times k_W}$, the convolution output $Y \in \mathbb{R}^{C_{\text{out}} \times H’ \times W’}$ is defined by:
>
> $$Y_{c, i, j} = \sum_{c'=1}^{C_{\text{in}}} \sum_{m=1}^{k_H} \sum_{n=1}^{k_W} K_{c, c', m, n} \cdot X_{c', i+m-1, j+n-1} + b_c$$
>
> This operation is equivalent to a sparse affine transformation and can be flattened into a set of linear constraints $Y = AX + b$, where $A$ encodes the convolution as a sparse matrix multiplication and $b$ is the bias. These constraints can be directly integrated into our MILP formulation, exactly like fully connected layers.
>
> **Our method also supports other non-linear activations (e.g., sigmoid, tanh)** using piecewise-linear upper and lower bounds. For instance, in the range [0, 1]:
>
> Sigmoid: $ 0.25x + 0.48 \leq \sigma(x) \leq 0.25x + 0.5$
>
> Tanh: $0.76x \leq \tanh(x) \leq x$
>
> Thank you for this suggestion! We will explicitly discuss these extensions in our revision.
>
> &nbsp;
> ### 3. Would it be possible to extend the evaluation to MNIST & CIFAR10 datasets.
>
> We thank the reviewer for this suggestion. While evaluation on MNIST and CIFAR-10 would indeed be valuable, **these benchmarks currently exceed the scalability** of our precise certification approach. Instead, we intentionally **focus on lower-dimensional datasets in safety- or security-critical domains**, which offer more **realistic and directly relevant application scenarios** for our method. Our first results on the suggestion by reviewer 4X7Z to restrict certified training to only the classification head look promising towards scaling to more complex models.
>
> &nbsp;
> ### 4. Please report also timing of FullCert and MIBP-Cert
>
> Thank you for this suggestion, we agree that a runtime comparison makes sense to illustrate the trade-offs involved. To address this, **we report the average runtime per epoch on the Two-Moons dataset**, in the same setting as Table 1. The detailed runtime of our method across epochs and perturbation radii is provided in Appendix C of our manuscript. FullCert and Sosnin et al. in pure IBP mode both take ~0.014s per epoch. IBP+CROWN mode (the default for TwoMoons) takes ~0.04s per epoch, and ~0.05s per epoch for CROWN. This is relatively consistent across different perturbation radii ($\epsilon$) and epochs. We report the average runtime for the first 5 epochs in the table below. Models typically either converge or diverge after 5 epochs.
>
> | Method | Time ($\epsilon=0.0001$) | CA ($\epsilon=0.0001$) | Time ($\epsilon=0.001$) | CA ($\epsilon=0.001$) | Time ($\epsilon=0.01$) | CA ($\epsilon=0.01$) |
> |---|---|---|---|---|---|---|
> | FullCert | 0.01s |83.9%|0.01s |82.2%|0.01s |71.5%|
> | Sosnin et al.| 0.04s |83.8%|0.04s |82.3%|0.03s |69.0%|
> | MIBP-Cert | 19s |83.3%|54s |82.0%|156s  |81.4%|
>
> While our method is slower, especially for larger $\epsilon$, it delivers significantly improved certified accuracy and training stability, validating our theoretical analysis regarding the runtime-precision trade-off. Furthermore, as Table 1 and Section 5.2 demonstrate, our method **uniquely certifies perturbation types that were previously impossible**. We will include an expanded runtime comparison in Section 5.3.
>
> &nbsp;
>
> Thank you again for your valuable feedback. We will incorporate it into our revised paper, which will significantly strengthen the submission.

---

> ### Comment · Area_Chair_goGT · 2025-08-06
> **Please read the rebuttal and discuss with the authors**
>
> Dear Reviewer mYxi,
>
> The paper has received mixed reviews, so your opinion is quite important. As the deadline for the discussion period approaches (August 8), please take a moment to read the authors' responses and reply to them at your earliest convenience. It’s important to give the authors ample time to respond, so please avoid waiting until the last minute.
>
> If you plan to maintain your current rating of the paper, kindly respond to the authors to confirm your support or any unresolved concerns. This feedback will be very helpful to the authors.
>
> Thank you,
> AC

---

> > ### Comment · Reviewer_mYxi · 2025-08-08
> >
> > Thank you for the comprehensive response and for addressing my earlier concerns. While I acknowledge that the presented method shows potential in terms of robustness certification, I find the improvements to be incremental and not substantial enough to constitute a decisive advantage. Therefore, I am maintaining my original score, as I remain unconvinced about recommending this work for acceptance.​​​​​​​​​​​​​​​​

---

> > > ### Author Response · Authors · 2025-08-09
> > >
> > > Dear Reviewer mYxi,
> > >
> > > Thank you for your follow-up. For clarity, we would like to briefly reiterate the main novel contributions of this work:
> > >
> > > - **New formulation** – first to cast training-time robustness certification as a mixed-integer bilinear program, enabling a different solution path.
> > > - **Theoretical stability** – we show that our approach addresses fundamental convergence issues in prior approaches.
> > > - **Expanded scope** – ability to certify perturbation types that no prior method can handle.
> > > - **Precision gains** – up to 20 pp certified accuracy improvement in challenging large-perturbation regimes.
> > >
> > > We see these as qualitative advances in methodology and scope, representing meaningful progress for robustness certification.

---

### Official Review · Reviewer_roE8 · 2025-06-30

**Clarity:** 3
**Significance:** 2
**Originality:** 3
**Rating:** 4
**Confidence:** 3

**Summary:**

This paper investigates the robustness verification of a neural network under potential data poisoning threats. The problem studied is essential. The authors proposes MIBP-Cert to provide robustness certificates. The paper is well-written and easy to follow. Please find detailed comments below.

**Questions:**

1. It is not clear why it is paramount to focus on ReLU MLPs.
2. The proposed method requires breaking down the NN into its trainable parameters and values for each neuron. It is unclear what would hinder scalability, depth, the number of neurons, or dimensionality. If time allows, the audience may benefit from adding a group of experiments to analyze the impact of these factors.
3. The authors claim the proposed method derives tighter bounds compared to bound-based methods. Does this claim still hold when the network gets deeper? If so, it would benefit the practical impact to make a comparison group to illustrate this trend.
4. Table 4 shows low certified accuracy. How does it compare to other approaches? Is this result showing the limitation of the proposed method? If so, please discuss potential countermeasures.

**Ethical Concerns:**

["NO or VERY MINOR ethics concerns only"]

**Final Justification:**

The authors have addressed my concern. However, given the scalability limitations of the proposed approach, I have decided to maintain my current score.

**Limitations:**

The primary limitation of the proposed method lies in its scalability. A more comprehensive experimental and analytical evaluation is needed to better assess how factors impact performance and runtime.

**Quality:**

3

**Strengths And Weaknesses:**

### Strength
1. The authors provide a comprehensive comparison in Section 2, using Table 1 to highlight the advantages of the proposed approach.
2. This paper is well-organized and well-written. Section 3 contains an illustrative example, making it easy to follow. Sections 3.2 and 3.3 present the proposed method in a clear and organized manner.

### Weakness
1. The proposed approach is limited to ReLU MLPs and does not apply to MLPs with other popular activation functions, such as sigmoid and tanh.
2. Potential scalability issue. Experiments focus on small MLPs (2 layers, 20 neurons per layer). The runtime of the optimization problem relies on the number of parameters that can grow exponentially with the size of the network.
3. Experiments show marginal improvement compared to existing works. Table 3 shows promising trends, but Table 3.b only indicates that the proposed approach has yielded a marginal improvement. Table 4 shows low certified accuracy. Table 5 shows the runtime but does not provide a baseline for comparison.

---

> ### Author Rebuttal · Authors · 2025-07-31
>
> Dear reviewer roE8,
>
> Thank you for your positive and constructive review! We address your concerns below; please let us know if anything remains unclear.
>
> &nbsp;
> ### 1. Why do you focus on ReLU activations instead of other choices such as sigmoid or tanh?
>
> ReLUs are among the most popular activation functions due to their efficiency and simplicity. Their piecewise-linear structure is ideal for precise certification, making them the preferred choice in most certification literature. **Solving the problem for ReLU is therefore the first step to piecewise linear bounding of general activation functions**. However, **our method supports other non-linear activations** (e.g., sigmoid, tanh) using piecewise-linear upper and lower bounds. For instance, in the range [0, 1]:
>
> Sigmoid: $ 0.25x + 0.48 \leq \sigma(x) \leq 0.25x + 0.5$
>
> Tanh: $0.76x \leq \tanh(x) \leq x$
>
> Thank you for this suggestion! We will explicitly discuss these extensions in our revision.
>
> &nbsp;
> ### 2.  Factors Affecting Scalability
>
> Theoretically, our complexity analysis in section 5.3 shows that the complexity bound is $O(2^{(n + k)b} \cdot 2p)$, meaning **the dominant factors are the number of activations $n$ and the number of classes $k$**. Therefore, worst-case scalability depends on these two values.
> To analyze real-world behaviour we run ablation studies on the model width and depth, which both influence the number of parameters and activations. We use the same models and data as in Table 1 with $\epsilon=0.01$, training for 1 epoch with varying network width and depth.
>
> | layers x width | total parameters | Training time |
> |---|---|---|
> | 2 x 5 | 27 | 0.75s |
> | 2 x 10 | 52 | 1.24s |
> | 2 x 20 | 102 | 4.79s |
> | 2 x 30 | 152 | 27.7s |
>
> | layers x width | total parameters | Training time |
> |---|---|---|
> | 2 x 5 | 27 | 0.75s |
> | 3 x 5 | 57 | 2.37s |
> | 4 x 5 | 87 | 6.24s |
> | 5 x 5 | 117 | 12.9s |
> | 6 x 5 | 147 | 22.1s |
>
> The numbers show that **training time corresponds roughly to the number of parameters**.
>
> &nbsp;
> ### 3. Are the bounds still tighter for deep networks?
>
> Yes, **our approach yields tighter bounds than previous methods for models of any size**, and this advantage grows with network depth. In deeper models, approximation errors from previous layers compound significantly, thus increasing the relative benefit of our tighter bound formulation. We will add this discussion and an experimental ablation to the revision.
>
> &nbsp;
> ### 4. Table 4 shows low certified accuracy. How does it compare to other approaches?
>
> Table 4 does **not** show low accuracy, but **our method's ability to model complex preconditions** that other methods cannot represent (complex perturbations in Table 1). The first line (“Assuming accurate health data”) shows the baseline accuracy without any perturbations or bounds, which is 56.4%, whereas the following lines show the worst-case impact of different data corruptions using our method. Missing mental health values have no measurable impact (same worst-case accuracy as the baseline), while the missing (unreported) values of all features could impact model performance by 2.8 percentage points (pp). Mental health over-reporting could be responsible for a 5.6pp drop in model accuracy.
>
> &nbsp;
> ### 5. Scalability limitations and potential for larger models
>
> This is **the first work that provides stable bounds for model training**, introducing a novel Mixed Integer Bilinear Programming formulation. We acknowledge that the scalability of our prototype implementation using an off-the-shelf solver is currently the main limitation, but demonstrate that it can be applied to a range of safety-, security-, or privacy-critical problems used in practice. We expect that follow-up work can improve on the scalability of our prototype implementation, similarly to how test-time certification methods have improved substantially over early, expensive methods that were limited to small model sizes as well. Potentially impactful directions include:
>
> - Hybrid methods combining our tight bounds selectively with faster approximations elsewhere
> - Exploiting model sparsity and structural properties for faster optimization
> - Exploring low-level engineering efficiency gains through optimized and/or parallelized implementations
> - Improvements in how to guide optimizer heuristics by exploiting domain/problem knowledge
>
> We will include this discussion in the revision.
>
> &nbsp;
> ### 6. Marginal Improvements and Baseline Comparisons
>
> Both Table 2 and Tables 3a-c show **consistent improvement in certified accuracy compared to the baselines**. This is especially noticeable for larger perturbation sizes, where **improvements are more than 11 percentage points (pp) on average, up to 20 pp** (10 pp (Table 2), 20pp (Table 3a), 4pp (Table 3b), 13pp (Table 3c)). These are meaningful improvements in the more challenging large perturbation bound settings.
>
> Furthermore, as Table 1 and Section 5.2 demonstrate, our method **uniquely certifies perturbation types that were previously impossible**. We will make this qualitative advantage even more explicit in the revised manuscript.
>
> &nbsp;
>
> Thank you again for your helpful feedback! We will incorporate it in the revision of our paper, which will significantly strengthen the submission.

---

> > ### Comment · Reviewer_roE8 · 2025-08-05
> >
> > We thank the authors for the detailed rebuttal. It addressed my concerns. However, given the scalability limitations of the proposed approach, I have decided to maintain my current score.

---

> > > ### Author Response · Authors · 2025-08-06
> > >
> > > Dear Reviewer roE8,
> > >
> > > Thank you for engaging with our rebuttal. We are glad to hear that it addressed your concerns.
> > >
> > > We acknowledge the scalability limitations of the current implementation. However, we respectfully suggest that the method’s novelty, correctness, training stability, and its ability to certify previously intractable perturbation types represent a meaningful and foundational contribution. As with many impactful works in robustness certification, initial methods often began with small models and higher runtimes, and scaled through sustained follow-up work [a, b, c]—a path we explicitly discuss and enable in our formulation.
> > >
> > > We are grateful for your positive assessment and believe that this work lays important groundwork for new directions in certified training.
> > >
> > > Thank you for your consideration.
> > >
> > > &nbsp;
> > >
> > > [a] Katz, Guy, et al. "Reluplex: An efficient SMT solver for verifying deep neural networks." International conference on computer aided verification. Cham: Springer International Publishing, 2017.
> > >
> > > [b] Tjeng, Vincent, Kai Y. Xiao, and Russ Tedrake. "Evaluating Robustness of Neural Networks with Mixed Integer Programming." International Conference on Learning Representations.
> > >
> > > [c] Brix, Christopher, et al. "The fifth international verification of neural networks competition (vnn-comp 2024): Summary and results." arXiv preprint arXiv:2412.19985 (2024).

---

### Official Review · Reviewer_4X7Z · 2025-07-03

**Clarity:** 4
**Significance:** 3
**Originality:** 3
**Rating:** 5
**Confidence:** 4

**Summary:**

This paper is an advanced form of training time certification. The idea is to try to certify a prediction as being valid regardless of a possible adversarial manipulation of the input data. During training, rather than performing normal SGD updates, parameters are updated and so are per parameter bounds that denote the possible range of the parameter given a fixed perturbation on the inputs. After training, these bounds are used during prediction, to provide predictions with certified ranges. The certification means that for any perturbation of the training data within the prescribed range, the prediction is guaranteed to lie within the calculated range. The difference to related works is that bilinear equations are established to compute the bounds, by analyzing the sums of parameters and activations from the previous layer, where the activation functions must be relu in order to yield tractable mixed integer objective, where the integer denotes whether the relu is active. The method gives tighter predictive bounds at the cost of more computation.

**Questions:**

Q1) isn’t the classic adversarial setting one in which the attacker perturbs a test point given a fixed trained model? Please comment on the general applicability of that and the proposed setting.
Q2) I feel like equation (1) could be put in a clearer way. (2) also seems kind of strange but maybe I’m just not familiar with this style of notation. I understand the meaning of both of course, so this is a style comment.
Q3) Does A get overloaded on lines 119 and 125 and equation (2)?
Q4) On line 127, what does it mean to solve (2)?
Q5) This may expose my naivety to the area - but could one optimize to convergence (a point in parameter space, no bounds), and then compute the bounds that occur by optimizing from that stationary point under a set of perturbed data? Could this somehow be more efficient that tracking from a random initialization?
Q6) How about pre-training a large model, and then performing certified post-training for just the prediction head assuming the attacker model? Would that just reduce to a single layer analysis that could be very efficient?
Q7) What is the number of branches? Please define this for non-specialists.

**Ethical Concerns:**

["NO or VERY MINOR ethics concerns only"]

**Final Justification:**

I have considered everything already, thanks.

**Limitations:**

yes

**Quality:**

3

**Strengths And Weaknesses:**

Strengths

This is an important problem. The proposed solution is highly non-trivial but the authors demonstrated practical improvements in terms of tighter bounds. It is remarkable that the authors made this work on real world problems (albeit small ones by today’s industrial ML standards). The paper includes a wonderful background and is very well explained, with thoughtful worked examples. I enjoyed reading the worked example, that shows how one obtains tighter bounds in a concrete example.

The paper gives stability guarantees where the related [18] actually discusses divergence of related methods, this is very cool!

Table 3 is great, and really makes the main point of the paper clear.

Weaknesses

To play devil’s advocate - the method is rather complex and presumably very slow to run. However, since the results are so strong, it is pareto optimal (for high compute budgets relative to dataset size). Hopefully, these ideas will pave the way for more efficient and related follow up papers.

It would be nice to see the timing for FullCert in the timing info in the appendix.

---

> ### Author Rebuttal · Authors · 2025-07-31
>
> Dear reviewer 4X7Z,
>
> Thank you very much for your thoughtful and constructive review! We are delighted that you enjoyed reading our paper and appreciated its relevance, impact, and clarity. Below, we address your questions and suggestions in detail.
>
> &nbsp;
> ### 1. Classic adversarial vs. training-time perturbations
>
> **The literature assumes different threat models**. Similarly to traditional software security, any input to a model can negatively influence its behavior. **People have mainly studied two sources: perturbation of the test data (e.g., traditional adversarial examples) and perturbation of the training data (e.g., poisoning attacks)**. While several certification methods exist for test data perturbations, we still largely lack such guarantees for training data, as demonstrated in our paper. Our method thus addresses this gap.
>
> Since training-time guarantees are a much more difficult problem, **it is straightforward to extend our guarantees to also include test-time perturbations**. We can change the post condition (eq. 2) to compute the certificate with respect to a (potentially) perturbed $\tilde{x}$:
> $$f_{\theta'}(\tilde{x}) = y \quad \forall \tilde{x} \in d(\tilde{x}, x) \land \forall D' \in \mathcal{D} \ \land\ \theta' = A(D', \theta_0).$$
>
> $d(\tilde{x}, x)$ defines the perturbation model of $x$, e.g. an $\ell_p$-norm of classical adversarial examples. We will discuss this potential extension in the revision of our paper.
>
> &nbsp;
> ### 2. Clarification of equations (1) and (2)
>
> Thank you! We appreciate the feedback and **will look into ways to make the equations more accessible**. Defining pre- and postconditions is common in the certification literature; we will aim to clarify it for a wider audience.
>
> &nbsp;
> ### 3. Overloading of Symbol $A$
>
> Yes, thank you for pointing this out! We will change the symbol to avoid confusion.
>
> &nbsp;
> ### 4. What does it mean to solve Eq. (2)?
>
> **We mean to verify if the equation holds**. We will clarify this in the revision.
>
> &nbsp;
> ### 5. Optimization from a stationary point rather than random initialization
>
> This is an intriguing idea! **It is not obvious to us how to arrive at valid bounds**, because the optimization recursively depends on the intermediate parameter values through the gradient descent steps (i.e., $\theta_i$ depends on $\theta_{i-1..0}$. But this could be an interesting idea to explore for future work.
>
> &nbsp;
> ### 6. Certified post-training for prediction head
>
> This is a great suggestion. Indeed, **certifying only the final prediction layer of a larger pre-trained model can substantially reduce complexity** and allow certified training of larger models. Inspired by your comment, **we are currently exploring experiments** in this direction, and will discuss this promising alternative in the revised manuscript.
> Preliminary **results are very promising**. We trained the same model as in Table 1 with $\epsilon$=0.01 on TwoMoons; once with normal, end-to end training, and once with a pretraining + finetuning setup. For the pretraining, we train on only 4 clean data points without perturbations. We then freeze all parameters except the last prediction head and continue training on the entire dataset with perturbation.
> | Training                  | Epochs       | Epoch Time                        | Total Time | Certified Accuracy |
> |--------------------------|--------------|-----------------------------------|------------|---------------------|
> | Full training            | 1            | 4.5 s                            | 4.5 s     | 80%                |
> | Pretraining + Finetuning | 2 pre + 1    | 0.35 s (pre), 0.7 s (fine)      | 1.4 s     | 81%                |
>
> Using this setup, we can also train larger models with at least 5 layers at the same width, which were previously out of reach. We believe this is a very promising direction and will include the results of our findings in the paper.
>
> &nbsp;
> ### 7. What is the number of branches?
>
> The number of branches refers to **the potential decisions a solver must explore in branch-and-bound methods** that are commonly used to solve MIBPs. Each binary variable creates two branches (one for each possible value). Although, theoretically, this could yield up to $2^n$ branches, in practice, solver optimizations and heuristics significantly reduce this number. To improve clarity, we will explicitly define this term and include a brief explanation in the revised paper.
>
> &nbsp;
>
> Once again, thank you for your valuable insights and suggestions, which we will incorporate in the revision of our paper.

---

> > ### Comment · Reviewer_4X7Z · 2025-08-04
> > **thanks for the clarifications**
> >
> > ...

---

> > > ### Author Response · Authors · 2025-08-06
> > >
> > > Thank you for your response and positive assessment! We appreciate your engagement and helpful suggestions, which we are incorporating into the revision.

---

### Note · Authors · 2025-08-14

Thank you very much for the constructive discussion.
### TLDR
We’re excited to introduce MIBP-Cert, the first method to cast training-time robustness certification as a Mixed Integer Bilinear Program (MIBP), **addressing prior work’s divergence**, and enabling **guarantees for richer, previously unattainable threat models**. Across datasets, it yields **tighter certificates** with substantial gains, up to 20pp. MIBP-Cert unlocks a principled, stable framework with clear routes to scale (hybrids, sparsity/structure, solver guidance/parallelization). In the rebuttal, we **added new ablations** (runtime, model architectures), **further layer types** (convolution, sigmoid, tanh), and a **scaling path via certified finetuning** (>3x faster). We are confident this is a strong, foundational contribution that can catalyze a new line of certified-training research.
### Novelty and significance
- **New formulation**: first to cast training-time certification as MIBP, enabling a precise, stable solution
- **Theoretical stability**: formal analysis that MIBP-Cert addresses fundamental convergence issues of prior work
- **Expanded scope**: ability to certify complex and expressive perturbation types (e.g., categorical features, structured/global constraints)
- **Precision gains**: exact bounds per training step
These improvements allow tighter certificates, stable training, and application in novel, safety-critical settings.
### Empirical gains
- **Tighter certificates**: consistently higher certified accuracy across all settings
- **Largest gains in hard regimes**: e.g., Iris +20pp, Breast Cancer +13pp
- **New threat models**: demonstrate complex perturbation types by quantifying the impact of data quality/reporting bias on medical data
### Additional evidence provided during rebuttal - as requested by the reviewers
- A promising scaling path: certifying only the prediction head for >3x speedup
- Ablations on model width/depth
- Expanded runtime comparisons
- New layer types: convolution, sigmoid, tanh
- Discussed joint training- and test-time certification
- Clarifications and notation improvements
### Scalability
We acknowledge higher compute: worst-case complexity inherits the nature of MIBP. As @4X7Z noted, in certification, early foundational methods are often complex, and then improve as they are optimized. Promising avenues include hybrid tight/relaxed regions, structure/sparsity exploitation, solver guidance/parallelization, and head-only certification.

---

### Decision · Program_Chairs · 2025-09-17

**Decision:**

Accept (poster)

**Comment:**

The paper aims to provide certification for training time data perturbations. The high-level insight is to compute a reachable set (bounds) of network parameters during training, and this set can be formulated and solved as a mixed integer bilinear program. During inference, these bounds are used to provide a prediction with certified guarantees, i.e., the prediction is guaranteed to be correct even in the presence of bounded training time data perturbations. Experiments were conducted mostly on small datasets and ReLU models, since solving MIBPs (using Gurobi) is quite expensive and restricted to ReLU only.

The AC believes the main contribution of the paper is its novel angle and formulation to tackle this problem. The scalability of MIBP is quite limited; however, as the first step in this direction, I believe this paper is worthy of publication, and the tight bounds provided by MIBP can serve as a baseline for future development of faster and relaxed methods. As future work, I encourage the authors to look into the recent progress of the neural network verification literature (check out approaches in VNN-COMP reports), where mixed integer programming has been replaced by more efficient methods such as bound propagation (not IBP), and achieve significantly better scalability. Additionally, there have been extensions of neural network verification beyond ReLU networks.